# Analysis of patients preferences in type 2 diabetes mellitus second-line drug treatment: A discrete choice experiment

Ann-Kathrin Fischer[ID][1,2]*, Andrew Sadler[2], Elke Mathey[3], Axel Mühlbacher[1,2]

1 Health Economics and Health Care Management, Hochschule Neubrandenburg, Neubrandenburg, Germany, 2 Gesellschaft für empirische Beratung mbH, An-Institut Hochschule Neubrandenburg, Berlin, Germany, 3 Novo Nordisk Pharma GmbH, Mainz, Germany

☉ These authors contributed equally to this work.
* akfischer@hs-nb.de

## Abstract

### Background

Type 2 diabetes mellitus (T2D) represents a major public health challenge with significant effects on morbidity and mortality. Clinical guidelines provide treatment recommendations, but there is limited understanding of patients' preferences. This study aimed to elicit preferences for second-line drug treatments for T2D.

### Method

A Discrete Choice Experiment with a partial-profile design was conducted from August to November 2023, involving German patients with experience in either monotherapy or second-line drug treatment. Participants completed 12 choice tasks, each presenting three alternatives described by attributes: risk of myocardial infarction, risk of stroke, risk of nerve damage, risk of nausea, risk of severe hypoglycemia, weight change, type and frequency of intake, and schedule of intake. Statistical analyses employed the Conditional Logit and Random Parameter Logit models to assess main effects and heterogeneity.

### Results

The study encompassed 583 adult individuals with T2D, evenly divided between the two populations. Key factors influencing choice decisions included risk of nausea, risk of nerve damage, and weight change, with weekly type and frequency of intake risk of myocardial infarction followed. Less impactful but still relevant were risks of stroke, and severe hypoglycemia, while the intake schedule was least significant. Analysis of BMI categories revealed distinct preferences, particularly in weight change, with significant heterogeneity observed among respondents.

**Data availability statement:** All relevant data for this study are publicly available from the Harvard Dataverse repository (https://doi.org/10.7910/DVN/D6VYNL).

**Funding:** This study was financially supported by Novo Nordisk Pharma GmbH (https://www.novonordisk.com) funding provided to independent research institute GEB mbH (Gesellschaft für empirische Beratung mbH) (http://www.empirische-beratung.de) and received by AF, AS, and AM. No additional external funding was received for this study. The funder had no role in study design, data collection and analysis, decision to publish, or preparation of the manuscript.

**Competing interests:** The authors have read the journal's policy and confirm there are no patents, products in development or marketed products associated with this research to declare. The authors also confirm adherence to PLOS ONE policies on sharing data and materials.

## Conclusion

This study highlights the importance of incorporating patient preferences into clinical decision-making. By quantifying preferences for second-line drug treatments, the study underscores the need for low-risk options that also consider weight change and intake conditions, aligning with the German National Health Care Guideline for T2D objectives for shared decision-making and treatment adherence. Recognizing individual sensitivities to risks and benefits is crucial for tailoring effective T2D treatment strategies. The study bridges clinical findings with patient perspectives, offering valuable insights into clinical practice, consideration for HTA processes, and design of clinical studies.

## Introduction

Type 2 diabetes mellitus (T2D) presents a significant public health challenge [1], compromising the functional capacities and overall quality of life. In Germany, around 7.2% of adults aged 18–79 have been diagnosed with diabetes mellitus, with T2D accounting for around 98% of the 4.6 million cases [2,3]. T2D leads to both microvascular and macrovascular complications, including heart disease, stroke, and neuropathy [4]. Therefore, achieving effective glycemic control is essential for managing the condition.

Unhealthy lifestyle factors contribute to an increased BMI, raising the risk of developing T2D [5,6]. Poor adherence to treatment is correlated with a significantly higher risk of complications [7,8]. Therefore, the first approach to T2D management involves lifestyle interventions. Drug interventions are only considered when non-drug approaches are insufficient, as recommended by the German National Health Care Guideline (NVL) for Type 2 Diabetes [9,10]. First-line interventions involve metformin monotherapy. For patients at cardiovascular risk or those not meeting glycemic targets with first-line interventions, a second-line drug treatment is recommended. Medications such as sulfonylureas (SU), dipeptidyl peptidase-4 inhibitors (DPP-4), glucagon-like peptide-1 receptor agonists (GLP-1), and sodium-glucose linked transporter 2 (SGLT2) inhibitors are added to the treatment regimen alongside metformin [10–12]. Treatment with SU is associated with an increased risk of hypoglycemia but offers enhanced glycemic control [13,14]. DPP-4 inhibitors have demonstrated lower efficacy in reducing glycemic parameters compared to GLP-1, SU and SGLT2 [15]. Furthermore, GLP-1 promote weight loss [16] but may cause gastrointestinal side effects [15]. In contrast, SU are linked to weight gain [14]. GLP-1 and SGLT2 are both associated with reduced risk of major adverse cardiovascular events [17].

These various treatment alternatives are associated with (1) different effects on clinical outcomes (e.g., glycemic control, cardiovascular risk), (2) side effects (e.g., gastrointestinal disorders, hypoglycemia), and (3) mode of administration (e.g., oral versus injectable). Ultimately, the decision on which treatment options to choose should be guided by patient preferences, which are essential for understanding the value of treatments [18].

The recent NVL for T2D emphasizes the importance of Shared Decision-Making (SDM), which has become the preferred model of interactions between healthcare providers and patients [10]. Evidence supports the efficacy of SDM in managing T2D, highlighting its role in enhancing treatment outcomes through patient involvement. Such approaches are essential for improving therapy adherence and patient outcomes [10].

## Aims and objectives

Health preferences research (HPR) significantly enhances SDM by identifying what matters most to patients. These studies systematically capture patients' values and priorities regarding treatment options, enabling healthcare providers to tailor recommendations to match individual preferences. By incorporating these insights, providers can offer personalized care that aligns with patients' goals, leading to more informed and satisfying decisions.

Furthermore, inadequate treatment adherence is significantly linked to a higher risk of complications [7]. Patient preferences are crucial in determining treatment adherence and achieving optimal outcomes. When treatments are in line with a patient's preferences, such as their choice of medication form, frequency, and potential side effects, patients are more likely to follow the prescribed regimen.

HPR is focused on understanding the value of treatments by eliciting preferences. The objective of this study was to gather scientifically robust evidence on patient preferences and to identify the relative importance of various aspects of second-line drug treatments for T2D. The aim is not only to identify patient preferences but also to integrate this knowledge into everyday clinical practice and the health technology assessment (HTA) process of medical products. This can inform the design of clinical studies and promote a more patient-centered approach to care, ensuring that treatment decisions align more closely with patients' values and needs.

## Materials and methods

Discrete Choice Experiments (DCE) are designed and conducted to quantitatively measure preferences [19,20], and are commonly used for assessing, ranking, and weighting attributes and levels of treatments based on their relative importance [21–26]. Rooted in economic theory and random utility theory, DCEs assume that participants make choices to maximize their utility, whereby revealing how they trade off attributes and levels of the options presented [20]. In this study, participants completed a series of choice tasks [27,28], each representing hypothetical alternative second-line drug treatments [29–31]. The design and application of this DCE followed established guidelines [32–36].

### Selection of attributes and levels

The inclusion of attributes was limited to those deemed essential for addressing the research question. The approach aimed to streamline the survey and ensure a focused and relevant assessment of T2D patients' preferences. Attributes and their levels were chosen based on specific criteria to ensure a targeted and relevant assessment of patients' preferences within the context of treatment decision-making. Firstly, only attributes with clearly defined preferences were included. Secondly, the attributes and levels were selected to accurately represent the framework (decision model) underlying treatment decisions for T2D. Thirdly, the selected attributes encompassed the full range of relevant clinical outcome parameters.

**Targeted literature research.** In January 2022, a focused literature review was conducted using PubMed. The search strategy consistently included the keywords "type 2" and "diabetes" combined with "metformin", "second anti-diabetic drugs" and favorable and unfavorable effects. This search identified 230 eligible reports, detailed in S1 Fig. The review identified 14 attributes associated with effects related to the reduction of T2D consequences (e.g., reduction of risk of stroke), 6 attributes linked to therapy-related effects (e.g., increased risk of gastrointestinal side effects), and 4 attributes concerning the administration mode (e.g., application of the drug). Further details are provided in S1 Table.

**Qualitative pilot interviews.** Qualitative semi-structured interviews were conducted with individuals diagnosed with T2D between October 12, 2022, and January 16, 2023. The objective was to identify the most important attributes. The participants were presented with 24 potential attributes and discussed various aspects including the consequences of their diagnosis, experiences with therapy-related side effects, drug administration, and limitations in their daily lives. Additionally, a ranking exercise using 24 cards was used to help participants express and prioritize the most important attributes. Each participant was asked to divide the attributes into "important" and "not important," and then to rank the most important ones. This enabled structured prioritization and required participants to make clear trade-off decisions.

A qualitative content analysis was conducted to evaluate the perceived relevance of each attribute. The findings from this analysis were triangulated with the individual rankings to identify which attributes were considered most decision relevant.

Some attributes, such as urinary tract infection, genital infection, risk of cardiovascular death, and hospitalization due to heart failure, were initially unfamiliar to participants in connection with T2D and its drug therapy. After presentation and expert discussions, these attributes were excluded due to their limited relevance to individuals with T2D.

**Decision model.** The German NVL was systematically decomposed into a structured decision framework (decision model), containing the most important attributes [10]. See S2 Fig. The model was developed based on two key assumptions: (1) glycemic control (HbA1c) serves as the primary indicator for reducing the risk of disease-related effects (T2D consequences), and (2) the therapy-related effects (treatment consequences) occur during the addition of second-line drugs. Attributes identified through the literature review and qualitative interviews were systematically mapped onto this framework, which categorized disease and treatment consequences into benefits and risks. The decision model was constructed to balance the trade-offs between diabetes-related complications (e.g., stroke) and therapy-related effects (e.g., nausea). These trade-offs were considered in relation to individual experience, disease stage, and sociodemographic factors. The highest ranked attributes were categorized accordingly, resulting in a final list of six attributes. In addition, two administrative attributes were included in the model. Attribute levels were derived from literature and clinical studies, reflecting the range of clinical outcomes. For attributes such as stroke risk and risk of severe hypoglycemia, the level differences were intentionally kept narrow to reflect the actual variation reported in contemporary clinical trials and approved treatment options at the time of study design. Detailed information can be found in S2 Table.

## Experimental design

The experimental design comprised 1,200 choice tasks blocked in 100 versions. Each respondent completed 12 choice tasks, each with 3 treatment alternatives. In addition to these tasks, respondents answered one dominance test (this task was not part of the experimental design) to assess the internal validity of their choices [37]. In the dominance test, one alternative clearly outperforms the others on all attributes, allowing researchers to check for consistent and attentive responding. A partial profile design was used for the DCE [38], featuring a total of 8 attributes. In each choice task, respondents were shown options which 5 attributes, with 2 attributes related to the application of the treatment (type and frequency of intake, schedule of intake) consistently present across all choice tasks. The remaining 6 attributes (risk of myocardial infarction, risk of stroke, risk of nerve damage, risk of nausea, risk of severe hypoglycemia, weight change) were rotated to ensure that each choice task systematically showed three of these attributes. The design aimed to balance survey complexity and data richness, minimizing respondent burden while capturing the key attributes that influence decision-making.

Attributes not included in the choice tasks were displayed above the choice tasks in the survey. These attributes had overlapping levels for all treatment alternatives, e.g., a risk of 0% for risk attributes and no change in body weight for the weight change attribute. The prompt above the choice tasks was: "Assuming that the attributes in the gray box are the same for all treatment options, which of the treatment options would you choose?" The use of visible level overlaps is intended to reduce dropout rates and improve attribute attendance in DCEs [39]. See S3 Fig.

## Survey instrument

Respondents first underwent an initial screening to confirm eligibility, which included questions about their consent, T2D diagnosis, current treatment, age, and German language proficiency. The web-based survey began with an introduction and explanations of second-line drug treatment for T2D. After providing detailed explanations of the attributes and levels used in the DCE, respondents answered questions about personal experiences, such as nerve damage or nausea.

Following a clear example of how the DCE would work, respondents proceeded to make their choices. The survey also collected personal information, therapy experiences, health status, quality of life, therapy adherence, and health information literacy. It concluded with an evaluation of the questionnaire. The survey was designed and administered using Sawtooth Software Lighthouse Studio 9.15.0. [40].

To ensure comprehensibility and usability, particularly with regard to medical terminology used in the attribute descriptions, pretest interviews using the Think-Aloud method were conducted with five individuals diagnosed with T2D. These interviews helped identify unclear or overly technical phrasing and informed subsequent refinements of the survey instrument.

## Study population and data collection

The study involved individuals diagnosed with T2D, divided into two groups. The first group included patients without prior experience with second-line drug treatment, currently receiving metformin monotherapy (in-experienced in second-line, Population 1). The second group included patients with experience in second-line drug treatment, currently undergoing combined treatment with metformin and an additional antidiabetic drug excluding insulin (experienced in second-line, Population 2). Inclusion criteria required participants to be 18 or older, residing in Germany, and capable of reading and understanding German. Data was collected from T2D patients in Germany, with potential participants identified through a professional healthcare market research provider using panels, self-help groups, and healthcare-related networks.. The quantitative data collection took place between July 17 and November 3, 2023.

## Statistical analysis

The analysis was based on the random utility theory (RUT), which assumes that individuals seek to maximize their utility when making decisions [41]. The utility ($U$) for individual ($i$) conditional on choice ($j$) is composed of an explainable or systematic component ($V_{ij}$) and non-explainable or random component ($\varepsilon_{ij}$): [42].

Choice data was analyzed using a conditional logit model (CL) and a random parameter logit (RPL) model. The CL provided an initial overview and tested convergence ability by verifying the expected direction of the coefficient signs [43,44]. In contrast to the conditional logit model, the RPL model accounted for the heterogeneity of preferences across the entire study sample and captured individual-specific variations [45–47].

Interpreting DCE results using regression models involves understanding coefficients, their significance, and implications. Coefficients represent preferences for attribute levels and their impact on choice probability. Standard error measures variability, smaller values indicate more precise estimates. A significant standard deviation in RPL suggested heterogenous preferences among respondents [45–47].

In an initial specification, all attribute-level parameters were modeled as random to account for preference heterogeneity. Based on model fit and statistical significance of estimated standard deviations, only parameters with significant heterogeneity were retained as random in the final model. All random parameters were assumed to follow a normal distribution; remaining parameters were modeled as fixed.

The choice data derived from the DCE was coded using effect coding, a technique similar to dummy coding. All attributes were modeled as categorical variables using effect coding to account for non-linear differences between levels. This method estimates the impact of each level relative to the overall mean of the attribute, rather than a single reference

category, and enables the identification of non-linear preference structures across attribute levels. In this approach, L – 1 variable are generated for each attribute, where L indicates the number of levels within that attribute [48]. Data analysis was performed using Stata 17.0 [49].

The analysis focused on main effects to determine the relative importance of treatment attributes and levels for therapy decisions. Interaction effects with sociodemographic characteristics were not part of the final model. Hypotheses were formulated for BMI, and related subgroup analyses were conducted. See Supporting Information S3–S5 Tables.

### Ethical approval

The study, including the suggested preference survey instruments, the informed consent form, and the study design, was reviewed and approved by the ethics committee of Hochschule Neubrandenburg (HSNB/195/22). In the qualitative study phases, all participants received a written participant information sheet prior to the interviews, along with informed consent forms. Both documents were reviewed and approved by the ethics committee. One copy of the informed consent form remained with the participant, while another copy was stored at the study center. For the quantitative study, a participant information document was made available for download, which was also reviewed and approved by the ethics committee. Participants provided their consent to participate at the beginning of the questionnaire through a consent question. If a participant declined, they were immediately excluded from the survey.

## Results

The data were collected during an online survey conducted between August and November 2023 among T2D patients in Germany. In total, there were 615 complete questionnaires. After data cleansing, the data of 583 adult patients (50% in each population) were used for the final analysis. Participants were excluded for straight-lining, consistently selecting two out of three options, failing validity tests, or completing the survey too quickly. There were 292 patients in population 1 and 291 in population 2. 55% of the total population were female and had an average age of 58 years. The majority were in partnerships or married (62.3%), had a secondary school leaving certificate (36.2%), a general qualification for university entrance (17.2%) or a university degree (13.4%). Most respondents were employed, full-time (42.5%) or part-time (25.7%), or were pensioned or retired (22.3%). See Table 1.

### Conditional logit model

An initial assessment of the plausibility of the data, the linearity of the parameters, the functional form of the variables and the predictability was carried out using a CL. Effect-coded data allows to calculate coefficients for each level of an attribute. The conditional logit model implicitly assumes that all respondents in the sample have the same preference structure and assumes a constant error variance across all respondents. Consequently, the model generates aggregated (mean) results that are consistent across all participants. Table 2 shows the analysis results of the conditional logit model. Most of the mean coefficients were significant at $p < 0.001$.

The attribute "Risk of nausea" had the highest mean coefficient, with the level "0 out of 100 patients (0%)" having a mean of 2.45. This suggests that participants strongly prefer treatment options with a lower risk of nausea. As the level of risk increases (e.g., 10%, 30%, 50%), the mean coefficients decreased significantly, indicating a strong aversion to higher levels of nausea risk. The attribute "Risk of nerve damage" seemed to be another key attribute influencing respondents' choices. The level "0 out of 100 patients (0%)" had the highest mean coefficient (1.48), also indicating a strong preference for T2D treatment options with no risk of nerve damage. As the level of risk increases (e.g., 5%, 10%, 15%), the mean coefficients decrease, signifying a negative impact on preferences. The attribute "Weight change" also influenced preferences, with a decrease of −6 kg and −2 kg having the highest mean coefficients (0.98 and 1.03, respectively). Respondents seemed to prefer T2D treatments that result in weight loss, while an increase of +2 kg and +6 kg has negative mean

**Table 1. Overview of current T2D treatment and sociodemographic data.**

| | Population 1, n = 292 | | Population 2, n = 291 | | Total, N = 583 | |
|---|---|---|---|---|---|---|
| **Current T2D treatment** | N | % | N | % | N | % |
| I only take one diabetes medication in tablet form (metformin). | 292 | 100 | 0 | 0 | 292 | 50.1 |
| I take a diabetes medication in tablet form (metformin) and also another diabetes medication in tablet form. | 0 | 0 | 189 | 65 | 189 | 32.4 |
| I take at least one diabetes medication (metformin) in tablet form and inject another diabetes medication (not insulin). | 0 | 0 | 102 | 35.1 | 102 | 17.5 |
| Patient characteristics | N | % | N | % | N | % |
| Age Mean – SD – Min – Max | 58 - 8.8 - 24 - 77 | | 57 - 8.6 - 29 - 74 | | 58 - 8.7 - 24 - 77 | |
| What is your gender? | | | | | | |
| Male | 123 | 42.1 | 136 | 46.7 | 259 | 44.3 |
| Female | 166 | 56.8 | 154 | 52.9 | 320 | 54.9 |
| Diverse | 3 | 1.0 | 1 | 0.3 | 4 | 0.7 |
| What is your marital status? | | | | | | |
| In a partnership or married | 173 | 59.3 | 190 | 65.3 | 363 | 62.3 |
| Single | 34 | 11.6 | 33 | 11.3 | 67 | 11.5 |
| Widowed | 15 | 5.1 | 6 | 2.1 | 21 | 3.6 |
| Divorced or separated | 70 | 23.9 | 62 | 21.3 | 132 | 22.6 |
| What is your highest level of education? | | | | | | |
| No school leaving certificate | 4 | 1.4 | 2 | 0.7 | 6 | 1.0 |
| Secondary school certificate | 34 | 11.6 | 34 | 11.7 | 68 | 11.7 |
| Secondary school leaving certificate | 111 | 38.0 | 100 | 34.4 | 211 | 36.2 |
| Technical college | 17 | 5.8 | 30 | 10.3 | 47 | 8.1 |
| General qualification for university entrance | 52 | 17.8 | 48 | 16.5 | 100 | 17.2 |
| Graduation from a technical college | 25 | 8.6 | 33 | 11.3 | 58 | 10.0 |
| University degree | 41 | 14.0 | 37 | 12.7 | 78 | 13.4 |
| Doctorate | 7 | 2.4 | 6 | 2.1 | 13 | 2.2 |
| Habilitation | 1 | 0.3 | 1 | 0.3 | 2 | 0.3 |
| Which best describes your current job status? | | | | | | |
| Employed full-time (more than 30 hours) | 111 | 38.0 | 137 | 47.1 | 248 | 42.5 |
| Employed, part-time (no more than 30 hours) | 82 | 28.1 | 68 | 23.4 | 150 | 25.7 |
| Self-employed or freelance | 23 | 7.9 | 14 | 4.8 | 37 | 6.4 |
| In training | 0 | 0.0 | 1 | 0.3 | 1 | 0.2 |
| Studying | 1 | 0.3 | 2 | 0.7 | 3 | 0.5 |
| Pensioned, retired or in partial retirement | 69 | 23.6 | 61 | 21.0 | 130 | 22.3 |
| Not working | 6 | 2.1 | 8 | 2.8 | 14 | 2.4 |
| Do you live in the countryside or in a city? | | | | | | |
| I live in the country | 125 | 42.8 | 98 | 33.7 | 223 | 38.3 |
| I live in the city | 167 | 57.2 | 193 | 66.3 | 360 | 61.8 |

values (−0.45 and −1.57, respectively), indicating a preference against weight gain. However, the respondents seemed to be indifferent between a weight decrease of 2 kg and 6 kg. The attribute "Risk of myocardial infarction" also showed significant variations in the mean coefficients. The level "0 out of 100 patients (0%)" had a positive mean of 0.72, suggesting a preference for options with no risk. As the level of risk increases (e.g., 2%, 4%, 7%), the mean values decrease, indicating a preference for lower risks. The coefficients of the attribute "Type and frequency of intake" showed preferences for a

**Table 2. Conditional logit model.**

| Attributes | Levels | Mean | se | t | p | [95% conf. inter.] | |
|---|---|---|---|---|---|---|---|
| Risk of myocardial infarction | 0 out of 100 patients (0%) | 0.715 | 0.041 | 17.500 | 0.000 | 0.635 | 0.795 |
| | 2 out of 100 patients (2%) | 0.176 | 0.042 | 4.220 | <0.001 | 0.094 | 0.258 |
| | 4 out of 100 patients (4%) | −0.243 | 0.043 | −5.620 | <0.001 | −0.327 | −0.158 |
| | 7 out of 100 patients (7%) | −0.649 | 0.046 | −14.070 | <0.001 | −0.739 | −0.558 |
| Risk of stroke | 0 out of 100 patients (0%) | 0.548 | 0.041 | 13.230 | <0.001 | 0.467 | 0.629 |
| | 1 out of 100 patients (1%) | 0.151 | 0.042 | 3.580 | <0.001 | 0.068 | 0.234 |
| | 2 out of 100 patients (2%) | −0.151 | 0.043 | −3.540 | <0.001 | −0.234 | −0.067 |
| | 4 out of 100 patients (4%) | −0.549 | 0.046 | −12.000 | <0.001 | −0.638 | −0.459 |
| Risk of nerve damage | 0 out of 100 patients (0%) | 1.483 | 0.044 | 33.480 | <0.001 | 1.396 | 1.570 |
| | 5 out of 100 patients (5%) | 0.289 | 0.042 | 6.800 | <0.001 | 0.206 | 0.372 |
| | 10 out of 100 patients (10%) | −0.190 | 0.044 | −4.350 | <0.001 | −0.276 | −0.105 |
| | 15 out of 100 patients (15%) | −1.581 | 0.060 | −26.350 | <0.001 | −1.699 | −1.464 |
| Risk of nausea | 0 out of 100 patients (0%) | 2.449 | 0.055 | 44.230 | <0.001 | 2.341 | 2.558 |
| | 10 out of 100 patients (10%) | 0.974 | 0.046 | 21.300 | <0.001 | 0.884 | 1.063 |
| | 30 out of 100 patients (30%) | −1.209 | 0.062 | −19.500 | <0.001 | −1.330 | −1.087 |
| | 50 out of 100 patients (50%) | −2.214 | 0.082 | −27.000 | <0.001 | −2.375 | −2.053 |
| Risk of severe hypoglycemia | 0 out of 100 patients (0%) | 0.580 | 0.041 | 14.290 | <0.001 | 0.500 | 0.659 |
| | 1 out of 100 patients (1%) | 0.114 | 0.041 | 2.760 | 0.006 | 0.033 | 0.196 |
| | 2 out of 100 patients (2%) | −0.185 | 0.043 | −4.280 | <0.001 | −0.270 | −0.100 |
| | 4 out of 100 patients (4%) | −0.509 | 0.046 | −11.050 | <0.001 | −0.599 | −0.419 |
| Weight change | Decrease of −6 kg | 0.977 | 0.043 | 22.790 | <0.001 | 0.893 | 1.060 |
| | Decrease of −2 kg | 1.034 | 0.042 | 24.810 | <0.001 | 0.952 | 1.115 |
| | Increase of +2 kg | −0.445 | 0.047 | −9.500 | <0.001 | −0.537 | −0.353 |
| | Increase of +6 kg | −1.565 | 0.060 | −26.190 | <0.001 | −1.682 | −1.448 |
| Type and frequency of intake | Oral 1x per week | 0.581 | 0.030 | 19.120 | <0.001 | 0.522 | 0.641 |
| | Oral 7 times a week | −0.182 | 0.032 | −5.730 | <0.001 | −0.244 | −0.120 |
| | Injection 1x a week | 0.350 | 0.030 | 11.510 | <0.001 | 0.290 | 0.410 |
| | Injection 7x a week | −0.749 | 0.035 | −21.450 | <0.001 | −0.818 | −0.681 |
| Schedule of intake | Independent of meals in the morning | 0.052 | 0.031 | 1.680 | 0.092 | −0.009 | 0.113 |
| | Dependent on meals in the morning | −0.115 | 0.032 | −3.630 | <0.001 | −0.176 | −0.053 |
| | Independent of meals in the evening | 0.037 | 0.031 | 1.200 | 0.232 | −0.024 | 0.097 |
| | Dependent on meals in the evening | 0.025 | 0.031 | 0.830 | 0.407 | −0.035 | 0.085 |

Log Likelihood (LL)= −4023.98; Degrees of freedom (df) = 24; Akaike information criterion (AIC)= 8095.96; Bayesian information criterion (BIC)= 8286.80; Mean = mean coefficient; se = standard error; p = p-value; conf. inter. = confidence interval.

once-weekly medication. "Oral 1x per week" had a positive mean of 0.58, "Injection 1x a week" also had a positive mean of 0.35. In contrast, "Oral 7x per week" (−0.18), and "Injection 7x a week" (−0.75) had a negative mean, suggesting a preference against frequent applications in a week. Participants showed a preference for once-weekly administration over daily administration, both for oral intake and, though to a lesser extent, for injections. They preferred once-weekly injections over taking a tablet seven times a week. The attribute "Risk of stroke" seemed to play a smaller role in the decision making, with the level "0 out of 100 patients (0%)" having a positive mean of 0.55, indicating a preference for options with no risk of stroke. As the level of risk increases (e.g., 1%, 2%, 4%), the mean coefficients decrease, suggesting a negative impact on preferences. The attribute "Risk of severe hypoglycemia" also shows a small impact on choice decisions with

preferences for lower risks, with the level "0 out of 100 patients (0%)" having a positive mean of 0.58. As the level of risk increases (e.g., 1%, 2%, 4%), the mean coefficients decrease, suggesting a preference for T2D treatment options with no or lower risk of severe hypoglycemia. The attribute "Schedule of intake" showed relatively small impact and small differences between the attribute levels. The smallest but statistically only significant level "Dependent on meals in the morning" had a negative coefficient (−0.12), indicating a slight preference against taking medication dependent on meals in the morning.

In Fig 1, a line graph shows the mean coefficients and their respective 95% confidence interval (CI) for each attribute level.

The relative importance of attributes was normalized on a 10-point-scale. The attributes "Risk of nausea", "Risk of nerve damage", and "Weight change" appeared to be the most important factors influencing decisions in the DCE. The remaining attributes also contributed to respondents' preferences, however, to varying degrees and with less influence.

The two populations were considered separately for further analysis. Comparing the results of both populations, most attributes showed similar patterns with minor deviations. However, in the attribute "Weight change" there was a noticeably higher mean coefficient for weight loss ("Decrease of 6kg") and a lower coefficient for weight gain ("Increase of 6kg") for population 2 compared to population 1. This indicated a potentially higher preference for weight loss and a stronger aversion to weight gain in population 2. Even if the confidence intervals for the levels "Decrease of 6kg" and "Decrease of 2kg" overlap, the trend showed a clear direction. Respondents in population 2 prefer to lose more weight than respondents in population group 1. See S3 Table.

Based on this, a more detailed analysis regarding the body mass index (BMI) was conducted. The sample was divided into specific BMI ranges: BMI < 25 (underweight or normal weight), BMI = 25–29.9 (overweight) and BMI >= 30 (obese). All respondents still preferred a decrease in weight to an increase in weight. Respondents with a BMI below 25 clearly preferred a lower weight loss of 2 kg (mean = 0.91, 95% CI [0.75, 1.07], p < 0.01) compared to a weight loss of 6 kg (mean = 0.57, 95% CI [0.41, 0.74], p < 0.01). Overweight people (BMI 25–29.9) seemed to be indifferent between a weight loss of 2 kg (mean = 1.09, 95% CI [0.99, 1.20], p < 0.01) and 6 kg (mean = 1.01, 95% CI [0.90, 1.12], p < 0.01). Obese people (BMI >=29.9) clearly prefer a weight loss of 6 kg (mean = 1.66, 95% CI [1.40, 1.91], p < 0.01) to a weight loss of 2 kg (mean = 1.14, 95% CI [0.91, 1.38], p < 0.01). See S4 Table.

### Random parameter logit model

CL model assumes that all respondents have uniform preferences and ignores possible differences in individual preferences. In contrast, the RPL model accounts for heterogeneity by considering a preference distribution within the study sample.

Overall, the mean coefficients in the RPL model led to a very similar result as in the CL model. However, large standard deviations in the data indicated significant heterogeneity between respondents or groups of respondents. See Table 3. Particularly when it comes to the attributes "weight change", "risk of nausea", "type and frequency of intake" and "risk of nerve damage", there may be respondents with different preferences and needs.

Fig 2 shows the mean coefficients of the random parameter logit model in the 95% confidence interval. In addition, the standard deviations of the random parameters are shown as dotted gray lines. The further the ends are apart, the greater the heterogeneity regarding this level. Here the heterogeneity within the population in terms of weight change becomes evident.

The relative importance of attributes from the RPL was similar to that from the CL.

The random parameter logit model was estimated separately for both populations. S5 Table presents mean coefficients and standard deviations of the attribute levels for the two populations. The attributes "weight change" and "type and frequency of application" notably showed significant variability in both populations, with diverse preferences for different levels. See S5 Table.

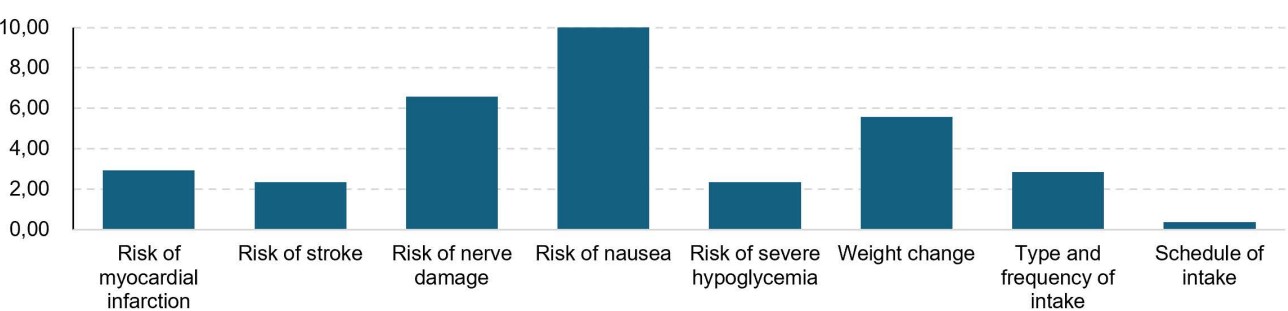

**Fig 1. Coefficients of the CL in 95% CI and relative importance.**

**Table 3. Random parameter logit model.**

| Attributes | Levels | Mean | se | p | [95% conf.] | | SD | se | p | [95% conf.] | |
|---|---|---|---|---|---|---|---|---|---|---|---|
| Risk of myocardial infarction | 0 out of 100 patients (0%) | 0.85 | 0.05 | 0.00 | 0.75 | 0.95 | . | . | . | . | . |
| | 2 out of 100 patients (2%) | 0.21 | 0.05 | 0.00 | 0.11 | 0.30 | . | . | . | . | . |
| | 4 out of 100 patients (4%) | −0.29 | 0.05 | 0.00 | −0.38 | −0.19 | . | . | . | . | . |
| | 7 out of 100 patients (7%) | −0.77 | 0.05 | 0.00 | −0.88 | −0.67 | . | . | . | . | . |
| Risk of stroke | 0 out of 100 patients (0%) | 0.66 | 0.05 | 0.00 | 0.56 | 0.75 | . | . | . | . | . |
| | 1 out of 100 patients (1%) | 0.19 | 0.05 | 0.00 | 0.09 | 0.28 | . | . | . | . | . |
| | 2 out of 100 patients (2%) | −0.18 | 0.05 | 0.00 | −0.28 | −0.09 | . | . | . | . | . |
| | 4 out of 100 patients (4%) | −0.66 | 0.05 | 0.00 | −0.76 | −0.55 | . | . | . | . | . |
| Risk of nerve damage | 0 out of 100 patients (0%) | 1.79 | 0.06 | 0.00 | 1.68 | 1.91 | 0.33 | 0.13 | 0.01 | 0.09 | 0.58 |
| | 5 out of 100 patients (5%) | 0.35 | 0.05 | 0.00 | 0.26 | 0.45 | 0.08 | 0.17 | 0.64 | −0.25 | 0.41 |
| | 10 out of 100 patients (10%) | −0.23 | 0.05 | 0.00 | −0.33 | −0.13 | −0.01 | 0.16 | 0.97 | −0.33 | 0.32 |
| | 15 out of 100 patients (15%) | −1.92 | 0.08 | 0.00 | −2.08 | −1.76 | −0.41 | 0.25 | 0.10 | −0.90 | 0.08 |
| Risk of nausea | 0 out of 100 patients (0%) | 2.95 | 0.09 | 0.00 | 2.77 | 3.12 | −0.40 | 0.14 | 0.00 | −0.67 | −0.14 |
| | 10 out of 100 patients (10%) | 1.18 | 0.06 | 0.00 | 1.05 | 1.30 | −0.19 | 0.12 | 0.11 | −0.43 | 0.04 |
| | 30 out of 100 patients (30%) | −1.42 | 0.08 | 0.00 | −1.58 | −1.27 | −0.30 | 0.35 | 0.38 | −0.99 | 0.38 |
| | 50 out of 100 patients (50%) | −2.70 | 0.12 | 0.00 | −2.93 | −2.47 | 0.90 | 0.35 | 0.01 | 0.21 | 1.59 |
| Risk of severe hypoglycemia | 0 out of 100 patients (0%) | 0.68 | 0.05 | 0.00 | 0.59 | 0.77 | . | . | . | . | . |
| | 1 out of 100 patients (1%) | 0.15 | 0.05 | 0.00 | 0.06 | 0.24 | . | . | . | . | . |
| | 2 out of 100 patients (2%) | −0.21 | 0.05 | 0.00 | −0.31 | −0.11 | . | . | . | . | . |
| | 4 out of 100 patients (4%) | −0.62 | 0.05 | 0.00 | −0.72 | −0.51 | . | . | . | . | . |
| Weight change | Decrease of −6 kg | 1.24 | 0.07 | 0.00 | 1.11 | 1.37 | 0.97 | 0.08 | 0.00 | 0.82 | 1.13 |
| | Decrease of −2 kg | 1.29 | 0.06 | 0.00 | 1.18 | 1.41 | 0.63 | 0.09 | 0.00 | 0.46 | 0.80 |
| | Increase of +2 kg | −0.41 | 0.06 | 0.00 | −0.53 | −0.30 | 0.34 | 0.14 | 0.02 | 0.07 | 0.62 |
| | Increase of +6 kg | −2.12 | 0.10 | 0.00 | −2.31 | −1.92 | −1.95 | 0.18 | 0.00 | −2.30 | −1.59 |
| Type and frequency of intake | Oral 1x per week | 0.72 | 0.04 | 0.00 | 0.63 | 0.81 | 0.61 | 0.05 | 0.00 | 0.51 | 0.71 |
| | Oral 7 times a week | −0.20 | 0.04 | 0.00 | −0.28 | −0.13 | −0.08 | 0.07 | 0.22 | −0.22 | 0.05 |
| | Injection 1x a week | 0.44 | 0.04 | 0.00 | 0.37 | 0.51 | 0.25 | 0.06 | 0.00 | 0.12 | 0.37 |
| | Injection 7x a week | −0.96 | 0.05 | 0.00 | −1.06 | −0.86 | −0.78 | 0.10 | 0.00 | −0.98 | −0.57 |
| Schedule of intake | Independent of meals in the morning | 0.06 | 0.04 | 0.13 | −0.02 | 0.13 | −0.10 | 0.07 | 0.13 | −0.24 | 0.03 |
| | Dependent on meals in the morning | −0.13 | 0.04 | 0.00 | −0.20 | −0.06 | −0.06 | 0.06 | 0.29 | −0.17 | 0.05 |
| | Independent of meals in the evening | 0.05 | 0.04 | 0.16 | −0.02 | 0.12 | 0.06 | 0.07 | 0.42 | −0.09 | 0.21 |
| | Dependent on meals in the evening | 0.02 | 0.04 | 0.49 | −0.05 | 0.09 | 0.10 | 0.12 | 0.40 | −0.14 | 0.34 |

Log Likelihood (LL)= −3874.775; Degrees of freedom (df) = 39; Akaike information criterion (AIC)= 7827.55; Bayesian information criterion (BIC)= 8137.666; Mean = mean coefficients; se = standard error; p = p-value; SD = standard deviation; conf. = confidence interval. The sign of the estimated standard deviations is irrelevant: interpret them as being positive.

## Discussion

This study presented a DCE investigating patient preferences for second-line drug treatment in T2D. The study design aimed to provide detailed insights into these preferences, facilitating the effective use of second-line drug treatments, particularly for patients with high cardiovascular risk or those for whom monotherapy has failed. The decision model was constructed to balance the trade-offs between diabetes-related complications (e.g., stroke) and therapy-related effects (e.g., nausea).

The statistical analysis highlighted that in the overall population "risk of nausea", "risk of nerve damage" and "weight change" were the most important factors influencing T2D treatment choices, followed by "type and frequency of intake"

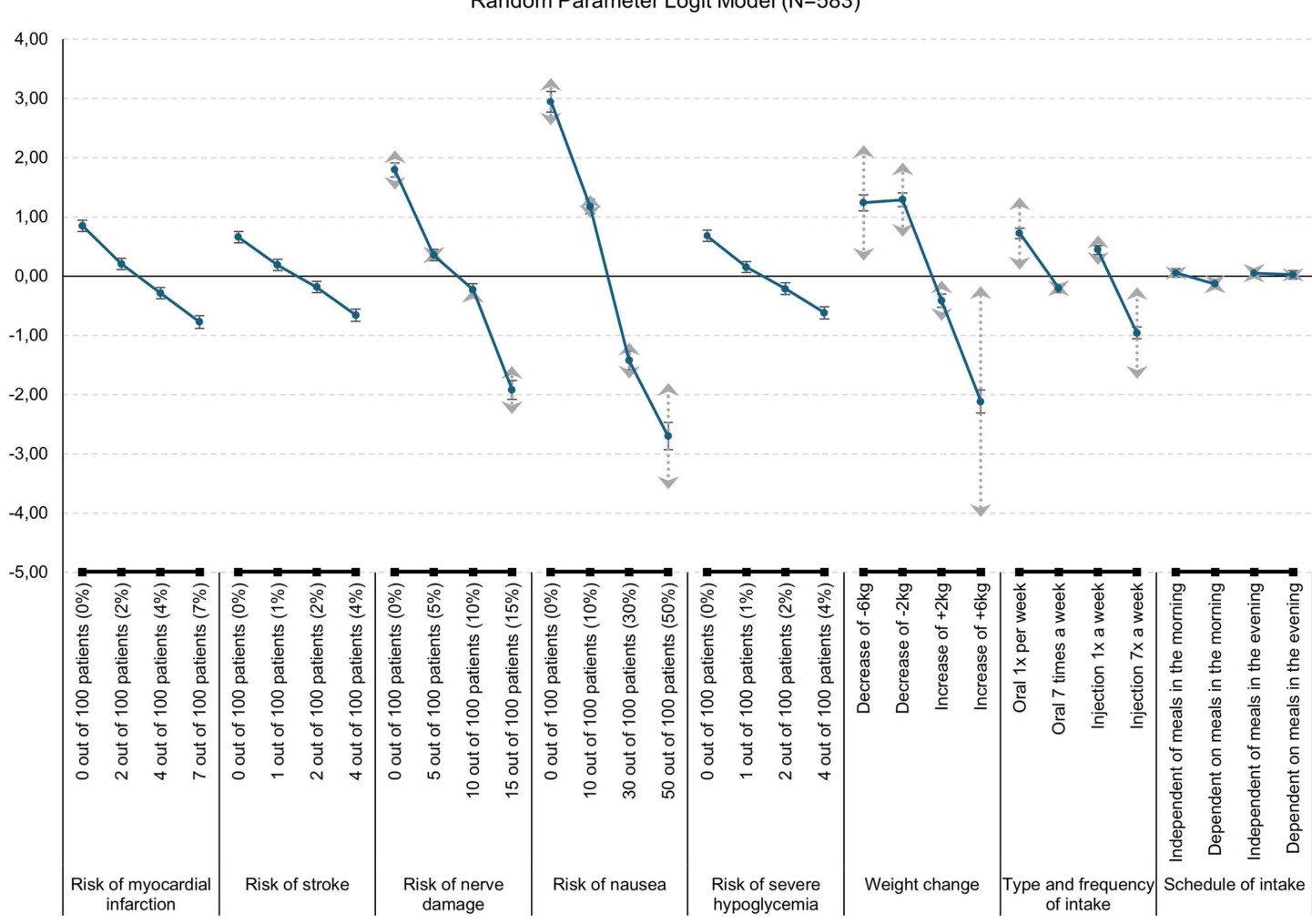

**Fig 2. Coefficients of the random parameter logit model in 95% confidence interval and standard deviations (grey arrows).**

and "risk of myocardial infarction". Respondents expressed a clear preference for treatments with a lower risk of nausea and nerve damage, as well as a preference for weight loss.

Further analyses using LC models indicated a stronger preference for weight loss in the population with prior experience of second-line treatments (population 2). The analysis of preferences by BMI category showed that respondents overall preferred weight loss overweight gain, though the desired extent weight loss varied by BMI.

While "weight change" was an important factor, "risk of nausea" emerged as the most important factor influencing choice decisions. The RPL model showed notable standard deviations, reflecting substantial heterogeneity in patient preferences, especially regarding attributes "weight change" and "type and frequency of intake".

Minimizing the risk of nausea and nerve damage is a key priority for T2D patients when evaluating treatment regimens. In addition, weight changes play a significant role from the patient's perspective, although sensitivity to weight changes varies across different individuals or groups. Lifestyle adjustments, dietary modifications, and exercise are essential components of T2D therapy, especially given that overweight and obesity often precede a T2D diagnosis [50]. Weight

reduction has therefore become a central target in selecting antidiabetic therapy. Insufficient glycemic control is linked to challenges in managing weight [51], increasing the likelihood of patients requiring second-line therapy.

Second-line therapy that promote weight loss is particularly favored for individuals with overweight or obese. Notably, individuals with a higher BMI seem to have a stronger desire for weight loss. These individuals are more likely to be categorized within Population 2 (those with second-line treatment experience), whereas those with a lower or no need for weight loss seem to belong to Population 1 (those with no prior second-line treatment experience). Qualitative interviews supported these findings, with most respondents expressing a general desire for weight reduction, although some respondents associated rapid or substantial weight loss with unhealthy conditions. The importance of weight change as a factor in decision-making seems to depend on patients' weight (BMI) and specific needs, such as controlling blood glucose levels.

Similar trends have been reported in other studies on T2D treatment preferences [52–56]. Despite the significant influence of the risk of myocardial infarction and stroke, these attributes appeared to exert less influence on respondents' choices compared to the risk of nausea, risk of nerve damage, and weight change. The higher likelihood of experiencing nausea or nerve damage, as opposed to the risk of a heart attack, may explain the stronger impact these attributes have on patient choices. It is also noteworthy that only 10.12% of the study population reported a diagnosis of neuropathy. While most participants had no personal experience with neuropathy, they still expressed strong negative preferences toward this condition. Upon further examination of the standard deviations RPL model, significant heterogeneity emerged, particularly in attributes such as weight change, risk of nausea, risk of nerve damage, and type/frequency of intake. These findings highlight the need for deeper exploration of this heterogeneity. To address this, latent class models (LCA) will be employed in subsequent analyses to identify and better understand the different preference patterns within the population. LCA will help reveal distinct subgroups based on shared preferences, allowing for a more tailored approach to T2D treatment [57,58].

The NVL emphasizes SDM as the preferred approach in patient-provider interactions, recommending the inclusion of individual patient perspectives. SDM improves treatment outcomes by enhancing patient involvement, adherence, and results. HPR helps identify patient priorities, enabling personalized recommendations. Integrating HPR into clinical practice improves care and reduces complications from poor adherence. Additionally, 83.88% of participants prefer shared decision-making with their healthcare providers, highlighting the importance of incorporating patient preferences for better alignment with values and improved satisfaction.

## Limitations

A limitation of this study is potential selection bias due to the small sample size (N = 10) in the qualitative interviews, which may affect the generalizability of the findings. To address this, a systematic approach was applied, including decomposing the German NVL T2D into a structured decision model with trade-off assumptions. Adjustments made after a pilot test were reflected in the final analysis with 30 respondents, as they did not significantly affect results or model coefficients. Partial profile design is used to reduce complexity when many attributes are involved, but it raises the issue of how to handle omitted attributes. One option is to hide them completely, reducing cognitive load, while another is to keep them visible but neutral, such as setting cost at a constant low level. Color coding can highlight overlaps but may introduce issues with readability. In this study, omitted attributes were displayed separately above each choice alternative to provide clarity while still giving respondents full information. Further research is needed to compare this approach with others [39,59].

The attribute levels were defined based on targeted literature research and supplemented by clinical studies to reflect real-world outcomes, using the minimum and maximum values documented in the literature. This systematic approach led to a greater disparity in the risks of nausea and nerve damage compared to myocardial infarction and stroke, as nerve damage is common in T2D patients [60], while the risk of nausea is mainly associated with the initial months of GLP-1 use [61]. Since nausea is time-restricted for GLP-1, the duration was not included in the attribute description, as nausea has

also been reported with other drugs without clear timeframes. The observed range of values may have influenced respondents' decision-making, potentially leading to a stronger focus on nausea and nerve damage. To address this potential bias, pretests incorporating the think-aloud technique were conducted between June 12 and June 15, 2023, to monitor and identify undesirable decision-making patterns. A specific scope test focusing on weight change was not conducted, but weight loss was indirectly addressed through the analysis of different BMI categories.

## Conclusions

This study demonstrates the importance of integrating patient preferences in the selection of second-line drug treatments. The findings emphasize the impact of nerve damage risk, nausea risk, and weight change in patients' treatment decisions. Patients strongly oppose side effects that affect daily life, particularly nausea. Furthermore, while weight change is a crucial factor, its importance varies across different subgroups. Respondents also showed a clear preference for weekly administration, whether oral or injectable, over daily treatment options.

Clinical decision-makers should prioritize these factors when transitioning patients from monotherapy to second-line therapy.

Tailoring treatment to individual preferences fosters a personalized, effective approach that improves health outcomes, adherence, and quality of life. This study offers valuable insights for clinical, policy, and regulatory stakeholders, potentially influencing drug approval, reimbursement, pricing, and treatment guidelines.

The objective of this study was to gather scientifically robust evidence on patient preferences and to identify the relative importance of various aspects of second-line drug treatments for T2D. The aim is not only to identify patient preferences but also to integrate this knowledge into everyday clinical practice and the health technology assessment process of medical products. This can inform the design of clinical studies and promote a more patient-centered care, ensuring that treatment decisions align more closely with patients' values and needs.

## Supporting information

**S1 Fig. PRISMA diagram.**
(DOCX)

**S2 Fig. Decision model.**
(DOCX)

**S3 Fig. Choice task, example.**
(DOCX)

**S1 Table. Attribute list.**
(DOCX)

**S2 Table. Descriptive framework.**
(DOCX)

**S3 Table. CL by study populations.**
(DOCX)

**S4 Table. CL by BMI categories.**
(DOCX)

**S5 Table. RPL by populations.**
(DOCX)

## Acknowledgments

The authors are grateful for the support of Blueberry Fields GmbH.

## Author contributions

**Conceptualization:** Ann-Kathrin Fischer, Elke Mathey, Axel Mühlbacher.

**Data curation:** Ann-Kathrin Fischer.

**Formal analysis:** Ann-Kathrin Fischer, Andrew Sadler, Axel Mühlbacher.

**Funding acquisition:** Axel Mühlbacher.

**Investigation:** Ann-Kathrin Fischer, Andrew Sadler, Elke Mathey, Axel Mühlbacher.

**Methodology:** Ann-Kathrin Fischer, Axel Mühlbacher.

**Project administration:** Ann-Kathrin Fischer, Axel Mühlbacher.

**Resources:** Axel Mühlbacher.

**Software:** Andrew Sadler.

**Supervision:** Axel Mühlbacher.

**Validation:** Ann-Kathrin Fischer, Andrew Sadler.

**Visualization:** Ann-Kathrin Fischer.

**Writing – original draft:** Ann-Kathrin Fischer.

**Writing – review & editing:** Ann-Kathrin Fischer, Andrew Sadler, Elke Mathey, Axel Mühlbacher.

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
