## [Decision Letter · Decision Letter 0]

1 Apr 2025

Dear Dr. Fischer,

Thank you for submitting your manuscript to PLOS ONE. After careful consideration, we feel that it has merit but does not fully meet PLOS ONE’s publication criteria as it currently stands. Therefore, we invite you to submit a revised version of the manuscript that addresses the points raised during the review process.

We look forward to receiving your revised manuscript.

Kind regards,

Hidetaka Hamasaki

Academic Editor

PLOS ONE

Journal Requirements:

This research is founded by the pharmaceutical company Novo Nordisk Pharma GmbH.

4. In the online submission form, you indicated that the datasets generated and/or analyzed during the study are not publicly available. The analysis will be published with the publication of the results. Individual data sets may be requested from the corresponding author upon reasonable request.

5. Please ensure that you refer to Figure 1 in your text as, if accepted, production will need this reference to link the reader to the figure.

Reviewers' comments:

Reviewer's Responses to Questions

**Comments to the Author**

1. Is the manuscript technically sound, and do the data support the conclusions?

Reviewer #1: Yes

Reviewer #2: Yes

2. Has the statistical analysis been performed appropriately and rigorously?

Reviewer #1: Yes

Reviewer #2: Yes

3. Have the authors made all data underlying the findings in their manuscript fully available?

Reviewer #1: Yes

Reviewer #2: Yes

4. Is the manuscript presented in an intelligible fashion and written in standard English?

Reviewer #1: Yes

Reviewer #2: Yes

Reviewer #1: A solid paper that could be enhanced by considering the analysis section, sub groups and the use of MRS.

In my opinion it would have been useful to have collected a cost attribute to enable calculation of willingness to pay.

Reviewer #2: Dear,

I hope this message finds you well. I would like to express my gratitude for the opportunity to review the manuscript titled "Analysis of Patients Preferences in Type 2 Diabetes Mellitus Second-Line Drug Treatment: A Discrete Choice Experiment." Overall, the paper is well-written and presents valuable insights into the topic. However, I have a few comments that I believe could enhance the clarity and depth of the study.

Firstly, I noticed that in the qualitative study section, there was no consultation with experts in the field of diabetes to complement the literature review. Given that physicians have direct interactions with patients, their insights could be incredibly beneficial to the findings.

Secondly, regarding the determination of layers for specific attributes, such as risk of stroke and risk of severe hypoglycemia, the differences between the levels appear to be quite trivial. It would be helpful to clarify the basis for these levels, especially since the differences between them are not tangible. This lack of substantial differentiation may explain why these two attributes showed the least impact in the results, as the distinctions do not seem significant or meaningful. I appreciate your consideration of these points, and I believe addressing them could strengthen the manuscript further.

**Do you want your identity to be public for this peer review?** For information about this choice, including consent withdrawal, please see our Privacy Policy

Reviewer #1: **Yes: ** Ann Livingstone

Reviewer #2: No

---

## [Author Response · Author response to Decision Letter 1]

28 Jun 2025

We sincerely thank you for the opportunity to revise and resubmit our manuscript. We are grateful for the constructive and thoughtful feedback provided by the reviewers and the Academic Editor. Your comments have been highly valuable in improving the clarity, methodological transparency, and overall quality of the manuscript.

In response to the editorial and reviewer comments, we have carefully revised our submission. Specifically:

We have prepared a detailed point-by-point response to all reviewer comments (see below).

We have uploaded a revised manuscript with tracked changes and a clean version.

As requested, we have added a clear citation to Figure 1 in the main text and ensured that all figures and tables are referenced appropriately.

We have also uploaded the full survey instrument in German and English as supplementary files for review only. The English version was generated using a standard, non-validated Microsoft Word translation feature for transparency. We emphasize that the survey instrument was thoroughly tested in German prior to data collection.

We have uploaded the supplementary material both with and without tracked changes, as the figure modifications occurred in that file.

For completeness, the following documents are included in this resubmission:

Revised manuscript with tracked changes

Clean version of the manuscript

Response to reviewers

Survey instrument (original German version)

Survey instrument (English translation)

Supplementary material with tracked changes

Clean supplementary material

Minimal anonymized dataset

We also confirm the following:

Funder Statement: The financial disclosure has been updated to reflect the required formulation:

“This research is founded by the pharmaceutical company Novo Nordisk Pharma GmbH. The funders had no role in study design, data collection and analysis, decision to publish, or preparation of the manuscript.” We have included this updated statement in the cover letter as requested.

Data Availability: After reviewing all ethical and legal considerations, we have decided to make the data publicly available. For the purpose of peer review, the minimal anonymized dataset has been uploaded via the submission portal. Upon acceptance, we will deposit the dataset in a stable public repository and provide the corresponding DOI or access link.

Formatting and References: We have carefully reviewed the manuscript to ensure compliance with the PLOS ONE formatting guidelines and we have reviewed the reference list for accuracy and completeness.

Reviewer #1

Comments

Answer

We sincerely thank Reviewer 1 for the careful and constructive review of our manuscript. Your comments were very helpful and showed great attention to methodological details. We especially appreciated your suggestions on how to improve the analysis section, the decision model, and the clarity of figures and tables. Based on your feedback, we revised several parts of the manuscript to make our methods and results more transparent and easier to follow. In the following, we respond to each of your comments in detail and describe the changes we made in response. All changes have been marked in the revised manuscript.

Reviewer #1: A solid paper that could be enhanced by considering the analysis section, sub groups and the use of MRS.

In my opinion it would have been useful to have collected a cost attribute to enable calculation of willingness to pay.

We sincerely thank the reviewers for their thoughtful and constructive comments on our manuscript. Their suggestions have significantly contributed to improving the clarity and overall quality of our work. We have carefully addressed all points raised and revised the manuscript accordingly. In the following, we provide a detailed, point-by-point response to each comment.

General Comments:

Regarding funding: I note the research was funded by the pharmaceutical company Novo Nordisk Pharma GmbH. Please confirm that they had no input into the attributes/levels and analysis of the DCE.

We confirm that Novo Nordisk Pharma GmbH, while providing financial support, had no influence on the selection of attributes and levels, nor on the design, conduct, or analysis of the Discrete Choice Experiment.

Abstract:

If the word count allows consider adding for type 2 diabetes at the end of the last sentence in the background: This study aimed to elicit preferences for second-line drug treatments for type 2 diabetes.

The abbreviation “T2D” was added at the end of the sentence to ensure consistency with the previous use of the term and to remain within the word count limit.

“This study aimed to elicit preferences for second-line drug treatments for T2D.”

Introduction:

Page 5, lines 57-59: The recent NVL for T2D emphasizes the importance of Shared Decision-Making (SDM), which has become the preferred model of interactions between healthcare providers and patients.

Please add a reference to this sentence.

A reference to the National Care Guideline (NVL) for T2D has been added to support the statement on the relevance of Shared Decision-Making.

Materials and Methods:

How were the 24 attributes identified in the literature reduced to 6?

On page 7, lines 120-2, there is some information concerning the following attributes being removed: “Some attributes, such as urinary tract infection, genital infection, risk of cardiovascular death, and hospitalization due to heart failure” However, this does not account for the processes of removal for the others? Please provide additional information about the processes used

The initial 24 attributes were presented and discussed in detail during the qualitative pilot interviews. Each participant evaluated the attributes individually and then ranked them using a structured card-sorting exercise. Attributes were first categorized as important or not important, and the most important were ranked to encourage explicit trade-offs. A qualitative content analysis was performed to assess the perceived relevance of each attribute. These results were compared with the rankings to identify those attributes most relevant for decision-making. We have revised the section on qualitative interviews accordingly to clarify the process used to reduce the attributes to a final set of six.

„Each participant was asked to divide the attributes into “important” and “not important,” and then to rank the most important ones. This enabled structured prioritization and required participants to make clear trade-off decisions.

A qualitative content analysis was conducted to evaluate the perceived relevance of each attribute. The findings from this analysis were triangulated with the individual rankings to identify which attributes were considered most decision relevant.”

Materials and Methods:

Page 8, lines 138-140, “Attribute levels were derived from literature and clinical studies, reflecting the range of clinical outcomes.”

No changes were made to this sentence, as no specific comment or request for revision was provided.

Materials and Methods:

Why was cost not considered an attribute? I’m assuming this varies considerably based on the type of drug, frequency of dosing, and mode of delivery.

Cost was not included as an attribute because the aim of the study was not to estimate willingness to pay (WTP), but rather to explore patients’ preferences regarding the clinical and experiential trade-offs between effectiveness, side effects, and treatment administration. Including cost could have introduced interpretive bias by shifting respondents’ attention toward monetary considerations, especially given the variability in individual out-of-pocket expenses in publicly funded healthcare systems. We therefore focused on attributes that directly reflect treatment-related benefits and risks from the patient perspective.

Experimental Design:

Page 8 lines 144-6, “In addition to these tasks, respondents answered one dominance test (this task was not part of the experimental design) to assess the internal validity of their choices [38].”

For the ease of the reader, I suggest a short sentence explaining the purpose of a dominance test.

A brief explanation of the purpose of the dominance test has been added to clarify that it is used to assess whether respondents make consistent choices when one alternative clearly dominates the others.

“In the dominance test, one alternative clearly outperforms the others on all attributes, allowing researchers to check for consistent and attentive responding.”

Experimental Design:

Did you pilot-test the survey or complete any think-aloud or cognitive interviews? Some of the text in the S2 Table – Descriptive Framework is heavy with medical terminology. I’m curious how easily understood it would have been. Did you use any methods to manage this issue?

While this aspect was briefly mentioned in the Discussion section, we agree that it deserves more prominence in the main text. We have therefore added a statement to the “Survey Instrument” section clarifying that pretest interviews were conducted to ensure comprehensibility and usability of the survey instrument, particularly regarding medical terminology used in the attribute descriptions.

“To ensure comprehensibility and usability, particularly with regard to medical terminology used in the attribute descriptions, pretest interviews using the Think-Aloud method were conducted with five individuals diagnosed with T2D. These interviews helped identify unclear or overly technical phrasing and informed subsequent refinements of the survey instrument.“

Survey Instrument:

Page 9, line 164, the paragraph stating: “Respondents first underwent an initial screening to confirm eligibility, which included questions about their consent, T2D diagnosis, current treatment, age, and German language proficiency. The web-based survey began with an introduction and explanations of second line drug treatment for T2D. After providing detailed explanations of the attributes and levels used in the DCE, respondents answered questions about personal experiences, such as nerve damage or nausea.”

Perhaps providing a copy of the survey instrument would help alleviate concerns around language and how easily understood the text/terms were for a medical condition and its health side-effects?

Thank you for your suggestion. To ensure full transparency regarding the linguistic and conceptual comprehensibility of the survey, we have provided the complete survey instrument as supplementary material for editorial and peer review purposes.

Specifically, we have included:

The original German version of the survey, which was rigorously pretested for clarity, comprehensibility, and content validity among individuals with type 2 diabetes prior to study launch.

An English translation of the survey generated using Microsoft Word’s built-in translation feature. While not a formally certified translation, it allows reviewers to navigate the content more easily while maintaining a direct reference to the original German source text.

Survey Instrument:

On page 10, lines 172-3, you state that the survey concluded with an evaluation of the questionnaire. What insights does that provide?

The evaluation provided insights into respondents’ perceived comprehensibility and acceptance of the questionnaire. A large majority (81.13%) found the questionnaire “easy” to complete, while 8.23% rated it as “very easy” and only 10.12% selected a neutral response (“teils-teils”); very few participants found it “difficult” (0.51%). Regarding future participation, 60.28% indicated they would “likely” fill out a similar questionnaire again, 22.80% answered “very likely,” and only a small minority expressed reluctance (2.59%). These results suggest high cognitive accessibility and broad acceptance of the survey instrument.

Study Population and Data Collection

Data was collected from T2D patients in Germany, with potential participants identified through a service provider.

Please provide additional details regarding how patients were identified. What service provider? Local Hospital, medical centre?

Participants were recruited through a professional healthcare market research service provider. The recruitment strategy included multiple channels such as panel respondents, individuals from disease-specific self-help groups, and outreach via healthcare-related networks. All participants met predefined eligibility criteria and confirmed a diagnosis of T2D before participation.

“Data was collected from T2D patients in Germany, with potential participants identified through a professional healthcare market research provider using panels, self-help groups, and healthcare-related networks.“

Statistical Analysis:

In my opinion several aspects of analysis are missing:

The functional form, were all attributes linear, or were other distributions tested and used?

Effects coding is discussed in the results but should be stated here

Were all parameters specified as random or non-random what was the process involved in this? How many parameters were random and non-random, and what were the distributions?

What socio-demographic characteristics were hypothesised a priori to have the potential of an interaction, e.g. gender, type of medication 1st or 2nd line, sex, BMI etc

What checklist was used to report the DCE? i.e. ESTIMATE, DIRECT please confirm and add the completed checklist as a table

Did you consider calculating Marginal rates of substitution to value trade-offs? If not, why not? I suggest doing so as another way to value the trade-offs

a. The functional form, were all attributes linear, or were other distributions tested and used?

All attributes were specified as categorical variables using effect-coding. This approach allows for the estimation of non-linear effects between attribute levels, and avoids assumptions of linearity across levels. No alternative functional forms were tested. The modeling approach reflects the assumption that preferences differ across distinct attribute levels and allows the analysis of the relative importance of levels and attributes. Our study aimed to capture discrete preferences across different levels. We added following sentence to the text.

“All attributes were modeled as categorical variables using effect coding to account for non-linear differences between levels.”

b. Effects coding is discussed in the results but should be stated here.

Yes, thank you very much. We have now added a short description of the coding method to Statistical Analysis section.

“All attributes were modeled as categorical variables using effect coding to account for non-linear differences between levels. This method estimates the impact of each level relative to the overall mean of the attribute, rather than a single reference category, and enables the identification of non-linear preference structures across attribute levels."

Bech, Mickael, and Dorte Gyrd‐Hansen. "Effects coding in discrete choice experiments." Health economics 14.10 (2005): 1079-1083.

c. Were all parameters specified as random or non-random what was the process involved in this? How many parameters were random and non-random, and what were the distributions?

Thank you very much! Originally, attributes were specified as random to allow maximum flexibility in capturing preference heterogeneity. In the final RPL model, only the attributes with statistically significant standard deviations were retained as random. The remaining attributes were treated as fixed. It was assumed that all random parameters follow a normal distribution.

“In an initial specification, all attribute-level parameters were modeled as random to account for preference heterogeneity. Based on model fit and statistical significance of estimated standard deviations, only parameters with significant heterogeneity were retained as random in the final model. All random parameters were assumed to follow a normal distribution; remaining parameters were modeled as fixed.”

d. What s

---

## [Editor Report · Decision Letter 1]

22 Jul 2025

Analysis of Patients Preferences in Type 2 Diabetes Mellitus Second-Line Drug Treatment: A Discrete Choice Experiment

PONE-D-25-09312R1

Dear Dr. Fischer,

We’re pleased to inform you that your manuscript has been judged scientifically suitable for publication and will be formally accepted for publication once it meets all outstanding technical requirements.

Kind regards,

Hidetaka Hamasaki

Academic Editor

PLOS ONE

Additional Editor Comments (optional):

Thank you for submitting the revised manuscript. I have reviewed it on behalf of the reviewers, and overall I believe you have addressed their comments and suggestions appropriately.
---

## [Editor Report · Acceptance letter]

PONE-D-25-09312R1

PLOS ONE

Dear Dr. Fischer,

I'm pleased to inform you that your manuscript has been deemed suitable for publication in PLOS ONE. Congratulations! Your manuscript is now being handed over to our production team.

Kind regards,

on behalf of

Dr. Hidetaka Hamasaki

Academic Editor

PLOS ONE